# Influence of Higher Order Viscous and Thermal Effects on an Ultrasonic Wave Reflected from the First Interface of a Porous Material

**DOI:** 10.3390/ma15030798

**Published:** 2022-01-21

**Authors:** Zine El Abiddine Fellah, Rémi Roncen, Nicholas O. Ongwen, Erick Ogam, Mohamed Fellah, Claude Depollier

**Affiliations:** 1Laboratory of Mechanics and Acoustics, French National Centre for Scientific Research LMA, CNRS, UMR 7031, Centrale Marseille, Aix-Marseille University CEDEX 20, F-13402 Marseille, France; ogam@lma.cnrs-mrs.fr; 2French Aerospace Lab, ONERA/Multi-Physics Department for Energy, Toulouse University, F-31055 Toulouse, France; Remi.Roncen@onera.fr; 3Department of Physics and Materials Science, Maseno University, Maseno 40105, Kenya; apache.nc.22@gmail.com; 4Laboratory of Theoretical Physics, Faculty of Physics, University of Science and Technology Houari-Boumediene BP 32 El Alia, Bab Ezzouar 16111, Algeria; fellahm1@yahoo.fr; 5French National Centre for Scientific Research, CNRS UMR 6613, Acoustics Laboratory of the University of Le Mans, UFR STS Avenue O. Messiaen, CEDEX 09, F-72085 Le Mans, France; Claude.Depollier@univ-lemans.fr

**Keywords:** porous materials, reflected wave, first interface, viscous and thermal surfaces, dynamic tortuosity and compressibility

## Abstract

Ultrasound propagation in porous materials involves some higher order physical parameters whose importance depends on the acoustic characteristics of the materials. This article concerns the study of the influence of two parameters recently introduced, namely, the viscous and thermal surfaces, on the acoustic wave reflected by the first interface of a porous material with a rigid structure. These two parameters describe the fluid/structure interactions in a porous medium during the propagation of the acoustic wave in the high-frequency regime. Both viscous and thermal surfaces are involved in Laurent expansion, which is limited to the dynamic tortuosity and compressibility to a higher order and corrects the visco-thermal losses. A sensitivity study is performed on the modulus of the reflection coefficient at the first interface as a function of frequency and on the waveforms reflected by the porous material in the time domain. The results of this study show that highly absorbent porous materials are the most sensitive to viscous and thermal surfaces, which makes the consideration of these two parameters paramount for the characterization of highly absorbent porous materials using the waves reflected from the first interface.

## 1. Introduction

Air-saturated porous materials such as plastic foams or fibrous materials are widely used in the field of sound insulation and passive-noise control [1]. Nowadays, noise pollution [2] is a public health problem [3,4,5] and acoustic comfort has become an important socio-economic issue [6]. In France, for example, noise pollution social costs 156 billion euros per year, according to the conclusions of a study by the French Noise Council and the Agency for Ecological Transition. Noise pollution has important consequences on health [4,7], well beyond hearing problems. It causes sleep disorders, of course, but also cardiovascular diseases, obesity and diabetes [7,8,9].

The prediction of acoustic attenuation in air-saturated porous media allows one to anticipate the control of noise pollution in the fields of housing [10], aeronautics [11], cars, urban transport [12,13,14,15,16] and public space [17]. Several methods have been developed for the characterization of air saturated porous materials [1,18,19,20,21,22,23,24,25,26,27,28,29,30,31,32] and more particularly the so-called acoustic methods, based on the experimental detection of waves transmitted and/or reflected by the porous material [33,34,35,36,37].

During acoustic propagation in porous material, different mechanisms of wave/ structure interaction take place [38,39,40]. These interactions are of viscous and thermal nature and are responsible for the sound attenuation [19,41,42,43]. Several physical parameters intrinsic to the porous material describe the architecture of the medium [1,38,39,44]. It all depends on the frequency: there are parameters that intervene in the high-frequency regime, such as the tortuosity and the viscous and thermal characteristic lengths [1,33,34,38,45], and parameters that intervene in the low-frequency regime, such as the viscous and thermal permeability [22,35,39], as well as other new parameters describing the inertia and the viscous exchanges in this frequency regime [44].

The high- and the low-frequency regimes are defined by comparing the thickness of viscous boundary layer with respect to the pore size [1]. When this layer is small compared to the pore size, we say that we are in the high-frequency regime [38,40] and when this layer is large compared with the pore size, we say that we are in the low-frequency regime [39,44]. The physics of the problem is not the same in high or low frequencies, indeed in each domain, the mechanical parameters describing the propagation and interactions between fluid and structure are different. To properly characterize the porous material using acoustic waves, it is necessary to look for the parameters in their frequency domain where they have the most influence on the propagation.

The regime of high frequencies corresponds to airborne ultrasound used for experimentation on porous materials saturated with air; generally, these frequencies range between 20 and 300 kHz [20,21,23,24]. Airborne transducers are used to send and detect acoustic waves propagating through the material and reflected by the interfaces of the medium [20,33,34].

Tortuosity and viscous and thermal characteristic lengths can be determined by solving an inverse problem, using the experimental data of the transmitted waves [46]. The use of reflected waves, and more precisely those reflected by the first interface of the porous material, makes it possible to obtain the porosity and the tortuosity [21,33,47,48]. The wave reflected by the first interface of the material is easily detectable experimentally because it is instantly reflected by the material and does not propagate in the medium and therefore does not undergo dispersion, being just attenuated by the reflection coefficient. It is then possible to measure in reflection the porosity and tortuosity with very good approximation. When the porous material is homogeneous, its physical properties are the same throughout the thickness of the material. The porosity and tortuosity measured at the first interface are the same throughout the thickness of the material. Recently, the viscous and thermal characteristic lengths have been measured experimentally using the waves reflected by the first interface of a porous material by solving the inverse problem using the Bayesian approach [49]. It has been shown that the measurement of these lengths is possible only for highly absorbent porous materials (having a large value of their specific resistance to the passage of the fluid) [49]. Indeed, these highly absorbent materials do not easily reach the regime of high frequencies contrary to low absorbent materials.

The viscous and thermal effects are described in two dynamic susceptibilities that describe the fluid and structure exchanges in the porous medium during the propagation of the acoustic wave [1,38,39,42]. The dynamic tortuosity describes the visco-inertial effects, and the dynamic compressibility describes the thermal exchanges. In each frequency domain (high and low), these two susceptibilities are expressed by a Laurent expansion as a function of frequency. The first-order expansion of these susceptibilities makes it possible to obtain the porosity and the tortuosity by the reflected wave [33,34,47,48], while the second-order expansion makes it possible to access the characteristic lengths [21]. Recently, two new parameters, the viscous and thermal surfaces, have been added for the third order expansion [40]. The use of the transmitted wave allowed for the measurement of the viscous surface.

The use of transmitted waves made it possible to highlight, on the one hand, the characteristic viscous and thermal lengths and, on the other hand, the viscous surface, with the sensitivity of the thermal surface being weak to the frequencies of airborne ultrasound in transmitted mode [49]. We propose, in this work, to see the effect of these viscous and thermal surfaces on the wave reflected by the first interface of the porous material, especially since the method using the wave reflected by the first interface has become a reference for the characterization of porous materials saturated with air in reflection, and it is thus interesting to see the influence of these surfaces on the reflected waves.

## 2. Model

The most general model describing the acoustics of porous media is the Biot model [50,51]. In this model, the acoustic wave propagates in the solid part as well as in the fluid part, with a strong coupling between the two phases. Generally, Biot’s theory is applied to a porous medium saturated by a heavy fluid such as a liquid for applications in geophysics [52] or for bone tissue characterization [53,54,55,56,57]. In porous materials saturated with air, the acoustic wave propagates essentially in the air saturating the material, with the structure remaining immobile, and in this case we use the equivalent fluid model [1,39], which is a special case of the Biot model.

In the high-frequency domain (20–300) kHz [38], which is defined when the viscous and thermal boundary layers $\delta ={\left(2\eta /\omega \rho \right)}^{1/2}$ and $\delta^{\prime }={\left(2\eta /\omega \rho P_{r}\right)}^{1/2}$ are small compared with the pore size *r*, the skin thicknesses become narrower and the viscous effects are concentrated in a small volume near the frame δ≪r and $\delta^{\prime }\ll r$.

Johnson et al. [38] described viscous effects in the dynamic tortuosity α(ω), and Allard [1] described thermal effects in the dynamic compressibility β(ω): (1)$$\begin{array}{ccc} & & \alpha \left(\omega \right)=\alpha_{\infty }+\frac{2\alpha_{\infty }}{\Lambda }{\left(\frac{\eta }{j\omega \rho }\right)}^{1/2}, \end{array}$$(2)$$\begin{array}{ccc} & & \beta \left(\omega \right)=1+\frac{2(\gamma -1)}{\Lambda^{\prime }}{\left(\frac{\eta }{j\omega \rho Pr}\right)}^{1/2}, \end{array}$$
where $\alpha_{\infty }$ is the tortuosity [58], Λ the viscous characteristic length, $\Lambda^{\prime }$ the thermal characteristic length, η the fluid (air) viscosity, ω angular frequency, γ the specific heat ratio and $P_{r}$ the Prandtl number. Some recent applications of this model are given in [59,60].

Later, Lafarge [39] extended the development to the higher order by adding the viscous and thermal surfaces in order to correct an extension already made by Pride [61] but which did not satisfy the particular condition of circular straight pores [40], which admits well known values for the tortuosity ($\alpha_{\infty }=1$) and the viscous and thermal characteristic lengths ($\Lambda =\Lambda^{\prime }=r)$, where *r* is the radius of the pore. The new extensions of the expressions of α(ω) and β(ω) are given by [40,49]: (3)$$\begin{array}{ccc} & & \alpha \left(\omega \right)=\alpha_{\infty }+\frac{2\alpha_{\infty }}{\Lambda }{\left(\frac{\eta }{j\omega \rho }\right)}^{1/2}+\frac{3\alpha_{\infty }}{\Sigma }\left(\frac{\eta }{j\omega \rho }\right), \end{array}$$(4)$$\begin{array}{ccc} & & \beta \left(\omega \right)=1+\frac{2(\gamma -1)}{\Lambda^{\prime }}{\left(\frac{\eta }{j\omega \rho Pr}\right)}^{1/2}+\frac{3(\gamma -1)}{\Sigma^{\prime }}\left(\frac{\eta }{j\omega \rho Pr}\right), \end{array}$$
where Σ and $\Sigma^{\prime }$ are the viscous and thermal surfaces introduced in Ref. [40].

### Reflection Coefficient

Consider a porous material modeled with the equivalent fluid model described in the previous section. This medium occupies the semi-infinite domain 0≤x≤∞. A sound pulse impinges on the medium from the left with an incident angle $\theta^{i}$. The geometry of the problem is given in Figure 1.

The wave reflected $p^{r}\left(x,t\right)$ from the first material interface is then given by:(5)$$\begin{array}{c} p^{r}\left(x,t\right)=\int_{0}^{t}\tilde{R}\left(\tau \right)p^{i}\left(x,t-\tau \right)d\tau , \end{array}$$
where $p^{i}$ is the incident wave and $\tilde{R}$ is the scattering reflection operator in the time domain. The reflection coefficient at the first interface R(ω) is calculated in the frequency domain for a porous material under oblique incidence, as a function of dynamic tortuosity and compressibility [21].
(6)$$\begin{array}{c} R\left(\omega \right)=\frac{1-E(\omega )}{1+E(\omega )}, \end{array}$$
where
(7)$$\begin{array}{c} E\left(\omega \right)=\frac{\phi }{\cos\theta_{i}}\sqrt{\frac{\beta (\omega )}{\alpha (\omega )}}\sqrt{1-\frac{\sin^{2}\theta_{i}}{\alpha (\omega )\beta (\omega )}}, \end{array}$$
with ϕ is the porosity, $\theta_{i}$ the incident angle, α(ω) the dynamic tortuosity and β(ω) the dynamic compressibility.

The reflection coefficient (Equation 6) is studied using the developments (Equation 1) and (Equation 2) of α(ω) and β(ω), where a detailed analysis on the influence of the different parameters (tortuosity, porosity, viscous and thermal characteristic lengths) has been performed [21]. It appears that porosity and tortuosity are the most influential parameters affecting the wave reflected by the first interface of the porous material, in other words, this wave is very sensitive to these two parameters. Indeed, within the range of the high frequencies [33,34,36,62]:(8)$$\begin{array}{c} R=\frac{\alpha_{\infty }\cos\theta -\phi \sqrt{\alpha_{\infty }-\sin^{2}\theta }}{\alpha_{\infty }\cos\theta +\phi \sqrt{\alpha_{\infty }-\sin^{2}\theta }}. \end{array}$$

Knowing the values of the reflection coefficient for two angles of incidence [33], it is then possible to calculate the porosity and tortuosity. This has been successfully applied for different porous materials saturated with air, such as plastic foams [33,47,48] and granular materials [63].

Recently [21], the second order of the limited development of dynamic tortuosity (Equation (Equation 1)) and compressibility (Equation (Equation 2)) has been used for the calculation of the reflection coefficient, making it possible to study its sensitivity towards viscous and thermal characteristic lengths, and it has been shown that the viscous characteristic length has a significant effect on the amplitude of the wave reflected by the first interface, especially for highly absorbent porous materials. The expression of the refection coefficient in this case is given by [21]:(9)$$\begin{array}{c} R\left(\omega \right)=\frac{1-E_{0}}{1+E_{0}}+\frac{2E_{0}\sqrt{\eta /\rho }}{\left(\alpha_{\infty }-\sin^{2}\theta \right){\left(1+E_{0}\right)}^{2}}\left(\frac{\alpha_{\infty }-2\sin^{2}\theta }{\Lambda }-\frac{\alpha_{\infty }\left(\gamma -1\right)}{\Lambda^{\prime }\sqrt{P_{r}}}\right)\frac{1}{\sqrt{j\omega }} \end{array}$$
with
(10)$$\begin{array}{c} E_{0}=\frac{\phi \sqrt{\alpha_{\infty }-\sin^{2}\theta }}{\alpha_{\infty }\cos\theta } \end{array}$$

The viscous characteristic length Λ is thus obtained by solving the inverse problem using the Bayesian approach on experimental signals reflected by real porous materials. However, the low sensitivity of the thermal characteristic length did not make it possible to determine it by inversion of the reflected experimental data [21].

In this work, we propose to study the sensitivity of the reflection coefficient and the waves reflected by the first interface of the porous material on the viscous and thermal surfaces Σ and $\Sigma^{\prime }$. For that we will use the Laurent expansions given by relations (Equation 3) and (Equation 4) in the general expression of the reflection coefficient (Equation 6) and (Equation 7).

## 3. Sensitivity Analysis

We now study the sensitivity of the viscous surface Σ and the thermal surface $\Sigma^{\prime }$ on the wave reflected by the first interface of the porous material. This sensitivity study allows us to know if the added parameters (surfaces) play a role in the attenuation of the waves reflected by the porous materials, and for which types of materials (very absorbent or low absorbent). Consider a highly absorbent porous medium M1 with the following physical properties: porosity ϕ=0.9, tortuosity $\alpha_{\infty }=1.1$, viscous characteristic length Λ=30μm, thermal characteristic length $\Lambda^{\prime }=45\;\mu$m, viscous surface Σ=90 pm${}^{2}$ and thermal surface $\Sigma^{\prime }=202.5$ pm${}^{2}$. The values of these parameters are characteristic of a highly absorbent porous material [22,33,34,42,46]. Generally, a porous material is highly absorbent when the values of its porosity as well as its viscous and thermal characteristic lengths are relatively low and the value of its tortuosity is high.

A realistic incident signal $p^{i}\left(x,t\right)$ is used to perform the numerical simulations and given in Figure 2; its spectrum in Figure 3. This signal is taken from our previous work published in Refs. [43,62].

Numerical simulations are obtained with the Matlab program by simulating the coefficient (Equation (Equation 6)), and the wave reflected by the first interface using the relationship (Equation (Equation 5)). Figure 4 shows the variation of the modulus of the reflection coefficient as a function of the frequency when the viscous and thermal surfaces are taken into account or not in the expressions of the dynamic tortuosity α(ω) and compressibility β(ω), for sample M1. The black curve corresponds to the case where the viscous and thermal surfaces are not taken into account, and the Johnson–Allard model is taken in its original form without any modification. In the red curve, only the viscous surface is considered, where the development of the dynamic tortuosity α(ω) is extended to the higher order corresponding to the viscous surface. In the blue curve, only the thermal surface is taken into account, where only the higher order term of the dynamic compressibility β(ω) is considered, with the viscous surface not being taken into account. Finally, the green curve corresponds to the case where both surfaces (viscous and thermal) are added in the development of α(ω) and β(ω).

We notice an important difference between the black and red curves, which indicates that the higher order, representing the viscous surface, is important in the modulus of the reflection coefficient. The comparison between the black and blue curves allows us to notice the effect of the thermal surface on the modulus of the reflection coefficient. Here, the change is also consequent, so these two quantities and the higher orders are important in this reflected mode. Finally, comparing the green curve with the reference curve in black allows us to conclude that taking into account both viscous and thermal surfaces simultaneously changes the modulus of the reflection coefficient significantly.

This same comparison is made on the waveforms reflected by the first interface of the porous material using relationship (Equation 5). It is interesting to see the effects on the waveforms, because it is the direct physical observable that the experimenter has on the oscilloscope before signal processing, especially since the signals are transient in the time domain. The same color codes are used to designate the influence of the viscous and thermal surfaces (Σ and $\Sigma^{\prime }$) on the reflected waves.

Figure 5 shows the effect of the surfaces Σ and $\Sigma^{\prime }$ on the waveforms reflected by the first porous material interface. Their spectra are given in Figure 6. We can see the reflected wave in black, obtained using the Johnson–Allard model without adding the surfaces Σ and $\Sigma^{\prime }$; in red, obtained by adding only the viscous surface Σ; in green, when only $\Sigma^{\prime }$ is added; and finally in blue, obtained when both surfaces are added.

The results presented in Figure 5 and Figure 6 show a real sensitivity of Σ and $\Sigma^{\prime }$ to the amplitudes and the attenuation of the reflected waves. These results are in the same direction as those given by the moduli of the reflection coefficients (Figure 4).

Consider now a low absorbent porous material M2, with the values of its physical properties being: ϕ=0.95, $\alpha_{\infty }=1.04$, Λ=100μm, $\lambda^{\prime }=150\;\mu$m, $\Sigma =1\times 10^{3}$ pm${}^{2}$ and $\Sigma^{\prime }=2.25\times 10^{3}$ pm${}^{2}$, and perform the same sensitivity study as that given below.

Figure 7 shows the variation of the modulus of the reflection coefficient as a function of the frequency for the second porous material M2. From this figure, we can see that the inclusion or exclusion of the viscous and thermal surfaces does not change much the shape of the curves. The curves are very close to each other, which means that whether or not taking into account the corrections of the dynamic tortuosity and compressibility by adding surface terms does not affect the reflection coefficient much. Figure 8 gives this same comparison but on the temporal waveforms of the signals reflected by the first interface of the porous material; we note here that the curves are almost identical. Their spectra given in Figure 9 shows almost no change among the curves.

This sensitivity study of the extensions in the Laurent expansion, in the high-frequency regime of the dynamic tortuosity and compressibility functions, on two types of porous materials, one highly absorbent and the other low absorbent, made it possible to highlight the effect of the viscous and thermal surfaces on the wave reflected by the first interface of the porous material. It appeared from the numerical simulations on the modulus of the reflection coefficient or on the reflected waveforms and their spectra that the highly absorbent porous materials are more sensitive to the viscous and thermal surfaces than the low absorbent materials. This result can be explained by the fact that highly absorbent materials reach the high-frequency regime characterized by inequality (δ≪r) less quickly than low absorbent materials. In this case, the correction added by the viscous and thermal surfaces is more significant for these highly absorbent materials.

## 4. Ultrasonic Measurement

Experiments were performed in air, using two Ultran NCT202 broadband transducers with a 190 kHz central frequency and a 6 dB bandwidth extending from 150 to 230 kHz. A Panametrics 5052PR pulser/receiver provided pulses with an amplitude of 400 V. A rigid metal plate was used to serve as an acoustic mirror to measure the wave emitted by the transducer. The signals received were amplified to 90 dB and filtered above 1 MHz to avoid high-frequency noise (low-pass filter). Electronic interference was removed by 1000 acquisition averages. The experimental setup is given in Figure 10.

A highly absorbent foam M3 saturated with air, previously studied and characterized with methods given in Refs. [24,46,62,64,65,66], was considered. The characteristics obtained from this foam were the following: porosity: ϕ=0.82, tortuosity $\alpha_{\infty }=1.7$, the viscous characteristic length Λ=23μm and the thermal characteristic length $\Lambda^{\prime }=3\Lambda$. This foam was characterized without taking into account the viscous and thermal surfaces Σ and $\Sigma^{\prime }$ added in this study. The incident experimental signal and the experimental signal reflected from the M3 sample are given in Figure 11.

By adjusting new values of porosity, tortuosity and characteristic length for the porous sample M3 and taking into account the viscous and thermal surfaces to the following values: ϕ=0.88, $\alpha_{\infty }=1.3$, Λ=33μm, $\Lambda^{\prime }=3\Lambda$, Σ=52.9 pm${}^{2}$ and $\Sigma^{\prime }=117$ pm${}^{2}$, we obtain Figure 12 with good experiment/theory agreement. The values of the parameters are adjusted manually to have a minimum of discrepancy between experiment and theory, while waiting to study in a future work the inverse problem numerically. We notice that there is a slight time difference between the experimental and theoretical signals, which could be minimized by solving the inverse problem in future work.

The extension of the dynamic tortuosity and compressibility model by adding the viscous and thermal surfaces Σ and $\Sigma^{\prime }$ in the modeling had a direct impact on the attenuation of the wave reflected by the first interface of the material; the consequence of the introduction of Σ and $\Sigma^{\prime }$ in the model is a different physical description of the interactions and visco-thermal exchanges between fluid and structure, which results in a modification in the characterization using the ultrasonic experimental data.

It is possible to characterize the porous material by solving the inverse problem using a simplified model with the least number of parameters without taking into account Σ and $\Sigma^{\prime }$, which have, however, shown their sensitivity for highly absorbent materials. However, the values obtained for the classical parameters, in this case for ϕ, $\alpha_{\infty }$, Λ and $\Lambda^{\prime }$, will be biased. This study has shown that these surfaces cannot be ignored for highly absorbent materials.

## 5. Discussion and Conclusions

Taking into account the viscous and thermal surfaces in the modeling results in the modification of the values of porosity, tortuosity and characteristic lengths during the characterization, and this is all the more true when the material is highly absorbent, as it was mentioned in the previous section. Now, what can be said about the different methods of characterization and inversion using experimental data? And where have these two surfaces been ignored? This work shows the importance of these two parameters Σ and $\Sigma^{\prime }$ on the attenuation of waves reflected by the first interface of the porous material. Neglecting these two parameters would eventually mean to overestimate or underestimate the classical parameters (porosity, tortuosity and characteristic lengths), which would lead to distort the characterization and the real values of these parameters.

The method of characterization based on the wave reflected by the first interface has several advantages, including the ease of experimental detection of this wave, as it is instantly reflected by the first interface. Moreover, this wave does not propagate in the porous material [21,33,34,47,48,67], so is not subjected to dispersion, only to attenuation. Remember that porosity is a difficult parameter to obtain by inversion of the experimental data in transmission, because the transmitted wave is not very sensitive to the porosity [67], while it is sensitive to the tortuosity and the viscous and thermal characteristic lengths. However, the wave reflected at the first interface is very sensitive to the porosity, hence the interest of experimentally exploiting this wave to trace the porosity, as well as the other parameters ($\alpha_{\infty }$, Λ and $\Lambda^{\prime }$). The wave reflected by the first interface of the porous medium has remarkable properties that the transmitted wave does not have, and therefore its use is essential for a complete characterization of the porous material.

Highly absorbent materials are those that best reflect the incident signal, and this reflection is proportional to their resistivity. It is thus easier to experimentally detect the reflected wave for highly absorbent media, but paradoxically, this study has shown that for these media, it is necessary to take into account the new parameters Σ and $\Sigma^{\prime }$. For good characterization of the material, it is essential to take into account these parameters, the importance of which this study has shown. The waves reflected by low resistivity materials are very attenuated and therefore more difficult to detect experimentally. Therefore, it is not necessary to consider viscous and thermal surfaces for the modeling and characterization of these materials.

This study allowed us to highlight the importance of the viscous surface Σ and the thermal surface $\Sigma^{\prime }$ in the attenuation of the waves reflected by the first interface, and it is now important to take into account these parameters in order to properly characterize a porous medium, especially since several methods of inversions have been developed in the literature to characterize porous media using the reflected waves, which have become essential thanks to their properties (attenuation and no dispersion) [21,33,34,47,48,67]. It is also necessary to repeat, in the future, the characterizations and inversions carried out on the highly absorbent porous materials to correct the reversed real values of the tortuosity $\alpha_{\infty }$ and of the characteristic lengths Λ and $\Lambda^{\prime }$.

## Figures and Tables

**Figure 1 materials-15-00798-f001:**
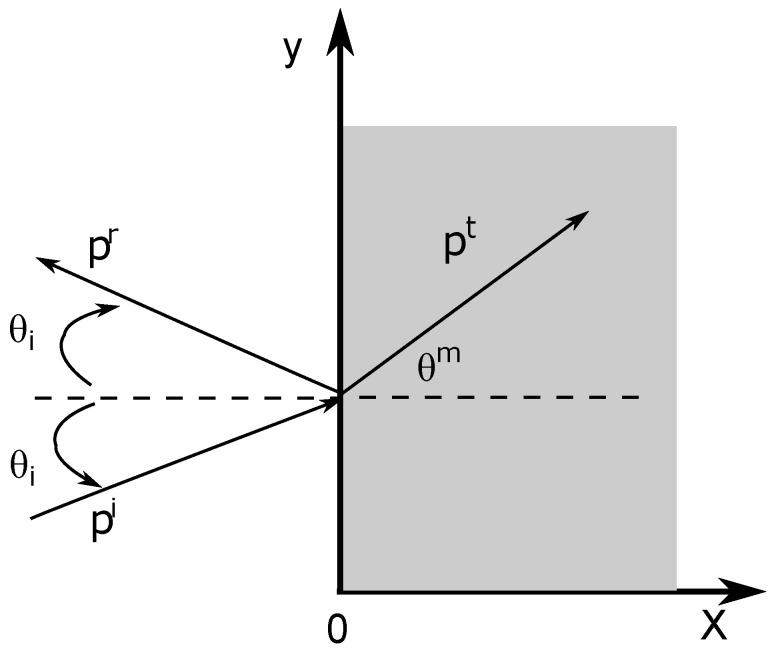
Geometry of the problem.

**Figure 2 materials-15-00798-f002:**
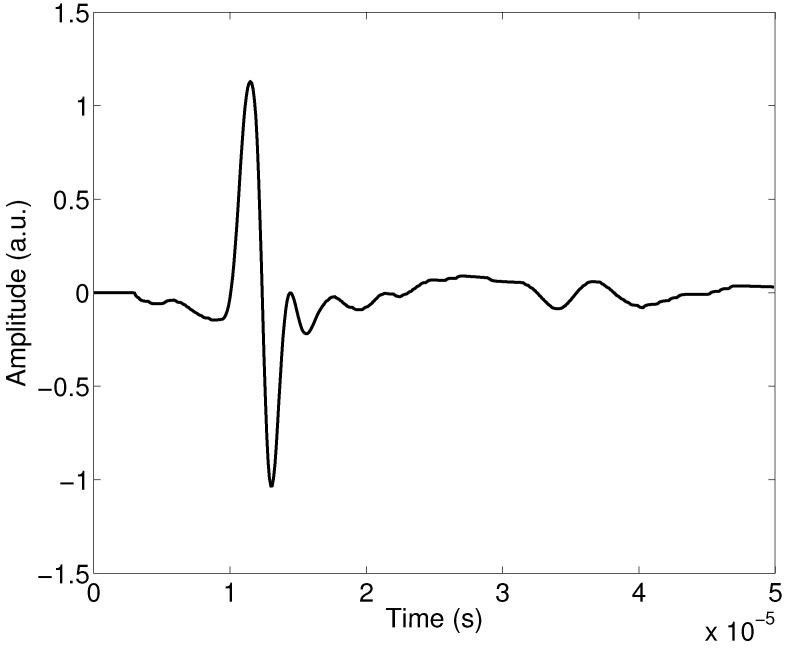
Incident signal used in the numerical simulations.

**Figure 3 materials-15-00798-f003:**
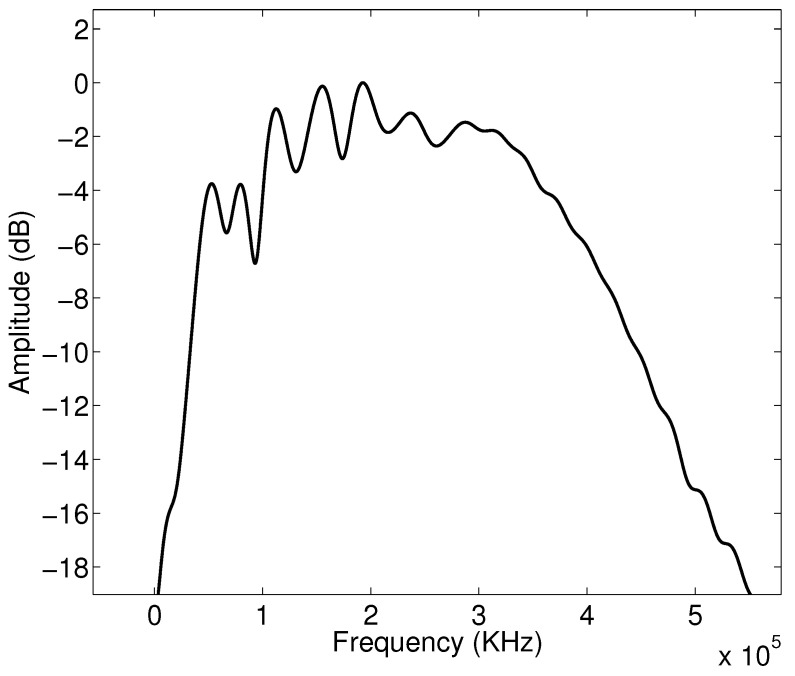
Spectrum of the incident signal.

**Figure 4 materials-15-00798-f004:**
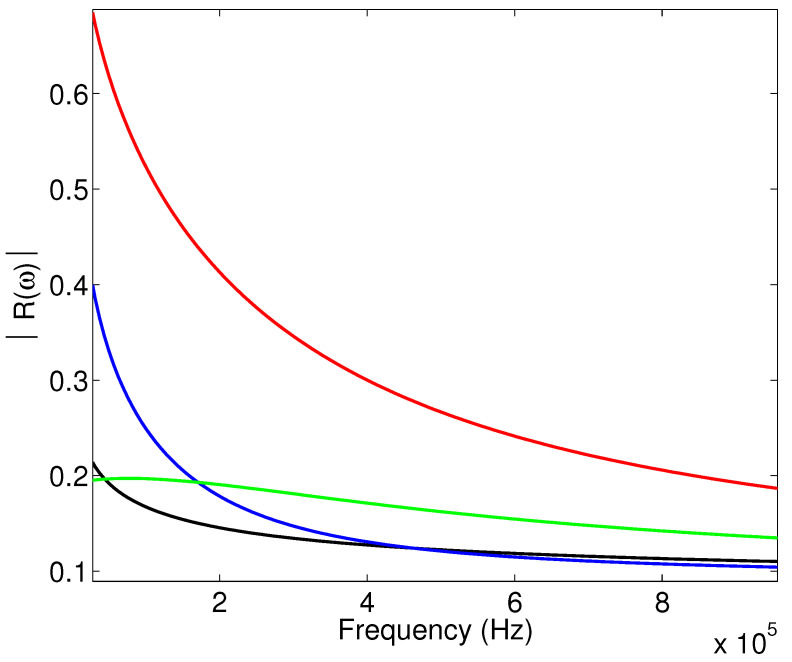
Variation of the modulus of the reflection coefficient as a function of frequency: study of the effect of viscous and thermal surfaces for sample M1.

**Figure 5 materials-15-00798-f005:**
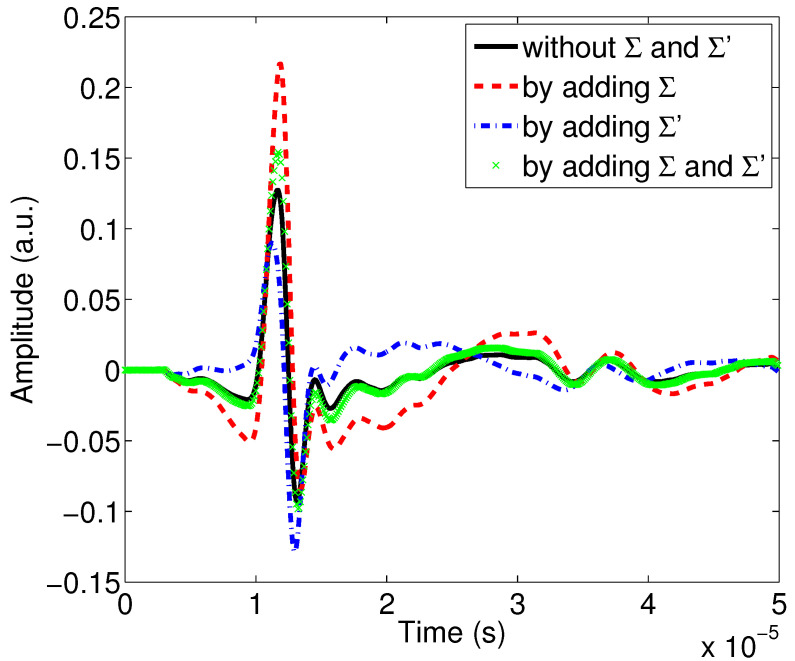
Study of the effect of viscous and thermal surfaces Σ and $\Sigma^{\prime }$ on the waves reflected by the first interface of the porous sample M1.

**Figure 6 materials-15-00798-f006:**
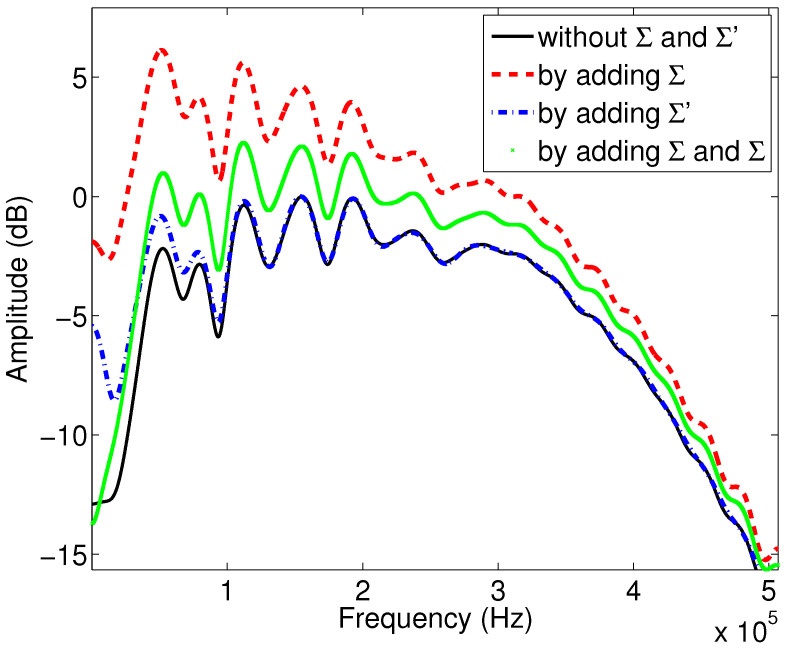
Study of the effect of viscous and thermal surfaces Σ and $\Sigma^{\prime }$ on the spectra of the waves reflected by the first interface of the porous sample M1.

**Figure 7 materials-15-00798-f007:**
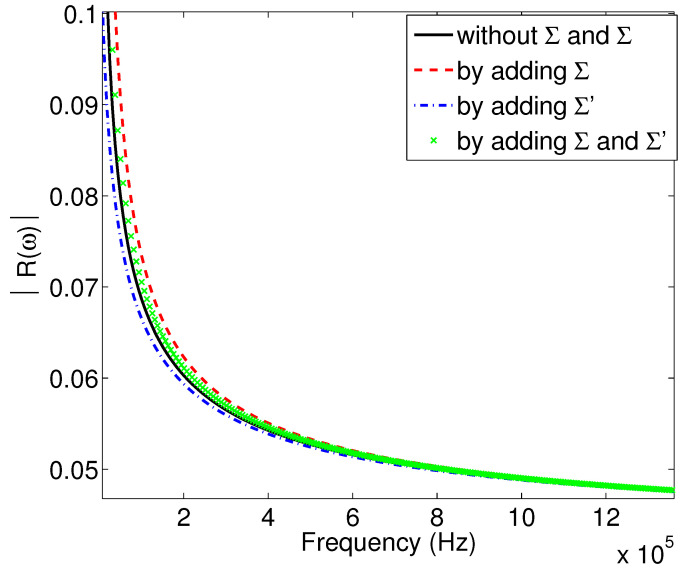
Variation of the modulus of the reflection coefficient as a function of frequency: study of the effect of viscous and thermal surfaces for sample M2.

**Figure 8 materials-15-00798-f008:**
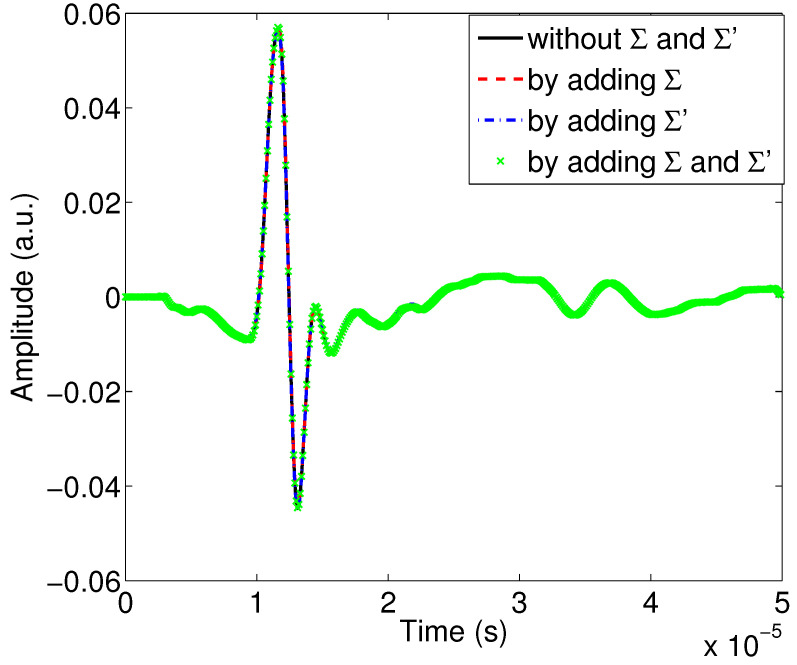
Study of the effect of viscous and thermal surfaces Σ and $\Sigma^{\prime }$ on the waves reflected by the first interface of the porous sample M2.

**Figure 9 materials-15-00798-f009:**
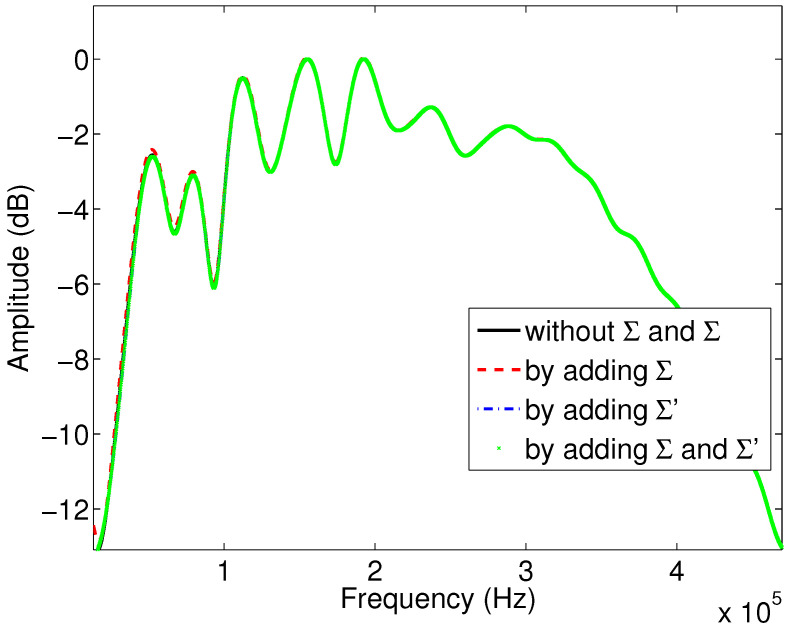
Study of the effect of viscous and thermal surfaces Σ and $\Sigma^{\prime }$ on the spectra of waves reflected by the first interface of the porous sample M2.

**Figure 10 materials-15-00798-f010:**
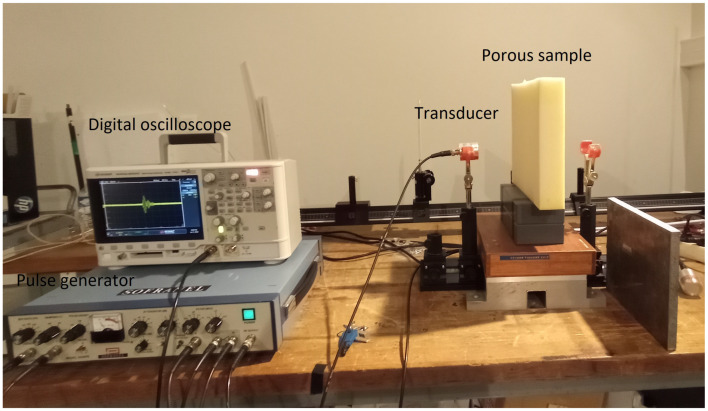
Experimental set-up of the ultrasonic measurement in reflection.

**Figure 11 materials-15-00798-f011:**
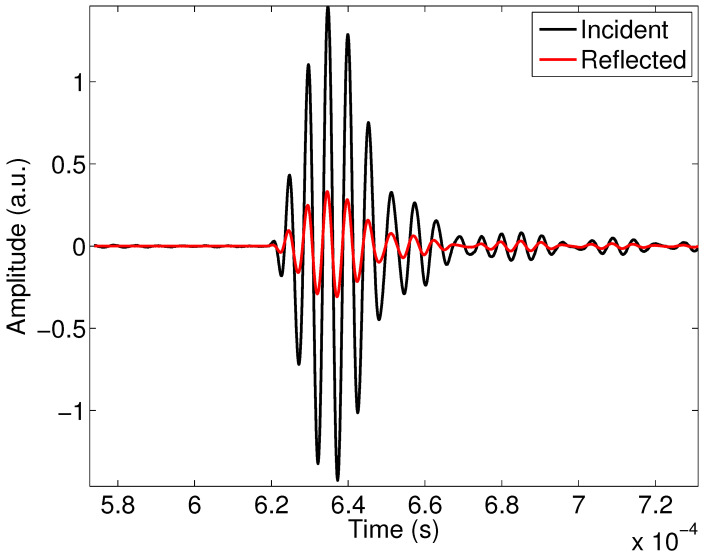
Incident and reflected experimental waves for sample M3.

**Figure 12 materials-15-00798-f012:**
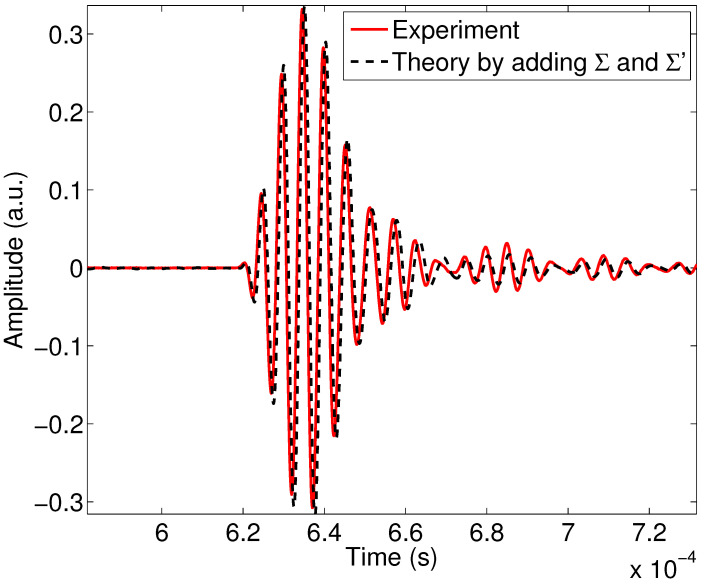
Experimental reflected wave (red line) and simulated wave (black dashed line) Σ and $\Sigma^{\prime }$ being taken into account.

## Data Availability

Not applicable.

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
