# Peer review of "Influence of Higher Order Viscous and Thermal Effects on an Ultrasonic Wave Reflected from the First Interface of a Porous Material"

_materials, 2022, doi:10.3390/ma15030798_

Round 1

Author Response

I would like to thank the reviewer for his advice and recommendations to improve our article.  We have responded to the comments point by point, correcting and improving the article according to the recommendations of the reviewer.

1/ Line 9, “The results of this study show that porous materials with a higher specific resistance to the passage of the fluid are the most sensitive to both viscous and thermal surfaces, in contrast to low resistive materials” ; this sentence has been changed to: « The results of this study show that the most absorbent porous materials are the most sensitive to viscous and thermal surfaces, in contrast to the least absorbent materials." the definition of "higher specific resistance" means "absorbent materials. Throughout the text the word "resistive" is replaced by "highly absorbent" and "non-resistive" by "low absorbent".

2/ Line 46, the frequency domain has been corrected in the text between 20 Khz and 300 kHz.

3/ Line 57.  When the porous material is homogeneous, its physical properties are the same throughout the thickness of the material. The porosity and tortuosity measured at the first interface are the same throughout the thickness of the material. This information has been added in the revised version of the paper.

4/ “the third order development” is replaced by “the third-order expansion” in the revised version. The purpose of this paragraph is to talk about what can bring the reflected waves but also transmitted, the paragraph is not made only for the reflected waves.

5/ Line 102, “w frequency pulsation” is replaced by should be angular frequency

6/Line 108. The character R is corrected in the text 

7/ α,β,θ are defined, respectively in  Line 122.

8/ Lines 157-160. Generally a porous material is highly absorbent when the values of its porosity as well as its viscous and thermal characteristic lengths are relatively low, while the value of its tortuosity is high. This information is added in the revised version of the paper.

9/ Line 161. We chose to use a realistic signal for the simulations in figure 2, this information is added in the text of the corrected version. It is quite normal that the signal in figure 5 has the same shape as the one in figure 2, since it is obtained by doing a simulation on the signal 2.

10/Line 163-164.  Numerical simulations are obtained with the Matlab program by simulating the coefficient (Eq. 6), and the wave reflected by the first interface using the relationship (Eq. 5), the incident signal (initial condition) is given by Fig. 2. This infirmation is added in the revised version of the paper.

11/ Lines 166-181, changes have been made to better explain the difference between the different curves (black, red, blue and green), as well as the scientific explanation.

12/ Figure 6 is corrected in the revised version.

13/ Line 189, the sentence « Moreover, on the waveforms we can observe the propagation of the wave and the attenuation of its amplitude », is deleted in the revised version.

14/Line 201, Σ’ = 2.25.103pm2 is revised as “Σ’ = 2.25×103pm2” in the revised version.

15/Line 211, “no change between the curves”, between is replaced with among in the revised version of the paper.

16/ Line 226, “Pulses of 400 V amplitude” is rephrased in the revised version.

17/ Line 229, “filtered above 1 MHz” means low-pass filter, this information is added in the revised version.

18/Line 229, an elegant figure is replaced with practical experimental equipment.

19/ The fonts in Figures 11 and 12 are corrected in the revised version.

20/ Lines 240-242. The values of the parameters are adjusted manually to have a minimum of discrepancy between experiment and theory, while waiting to study in a future work the inverse problem numerically. This information has been added in the revised version of the paper.

21/ Lines 236, 241. The incident experimental signal and the experimental signal reflected from the M3 sample are given in Figure 11. The comparison between experiment and theory is given in figure 12. This is corrected in the revised version of the paper.

22/The sentence «“the consequence of the introduction of Σ and Σ’ in the model is the change of the values of porosity, tortuosity and viscous and thermal characteristic lengths in the ultrasonic characterization of the porous material” in the old version is replaced by the sentence «  the consequence of the introduction of $\Sigma$ and $\Sigma'$ in the model is a different physical description of the interactions and visco-thermal exchanges between fluid and structure, which results in a modification in the characterization using the ultrasonic experimental data ».

23/Line 261, the sentence « this is all the more visible » is replaced by « this is all the more true ».

24/Lines 262-264. The sentence « Now, what can be said about the different characterization methods that have been developed in the past based on the expressions of dynamic tortuosity and compressibility, and about the different approaches to solving the inverse problem using experimental data? And where were these two surfaces ignored? » is replaced  in the revised version  by « Now, what can be said about the different methods of characterization and inversion using experimental data? And where have these two surfaces been ignored ».

25/Line 272. The wave reflected by the first interface is instantly reflected by the material and does not propagate in the porous medium. This instantaneous reflection results in an attenuation of the wave without dispersion, this result has already been demonstrated in our previous work in Refs. 24,25,40,41, 53-55, and as the medium is homogeneous, the porosity measured at the surface of the interface is the same as in the porous medium. This citation is added in the corrected version of the article.

26/Line 275. For porous materials with rigid structure and using the equivalent fluid model as in this work, the transmitted wave is not sensitive to the porosity in ultrasound, and the determination of the porosity using the ultrasonic transmitted wave is very difficult, precisely because of this low sensitivity, and this has already been shown in our previous work Ref 58. This reference is added and quoted in the corrected version of this article.

27/Line 296 ; “thanks to their properties” which  are « attenuation with no dispersion », this information is added in the revised version with the adequates citations.

Reviewer 2 Report

The article is well written. The Introduction includes documented literature. I propose some minor additions (see below). 
In your Introduction, you write about the health impact of noise and the negative impact of road noise. I propose to complete the literature in this regard, for example:

  • Applied Acoustics (2020), 159, article number 107080
  • Acoust. Soc, Am (2012), 132(6), 3788-3808
  • Environ, Health Perspect (2012), 120(1), 50-55

Author Response

We thank you for your proposals and suggestions for the improvement of the article, we have taken them into account by adding the references requested.

Reviewer 3 Report

  • Title is clear and reflects the content of the paper.Delete the full stop.
  • The abstract should be improved. Background should be added to place the question addressed in a broad context and highlight the purpose of the study.
  • Introduction/problem statement (Line 39-48) in the abstract need to be stronger, please revise.
  • It is recommended to enrich the introduction with the porousmaterials.
  • Discussion and Conclusions are sufficient.
  • Please add the photo of the experimental set-up.
  • It is recommended to write the conclusion in point form. Focus on the mean points.

Author Response

We thank you for your comments and suggestions to improve our article.

1/The stop point is removed at the end of the title as requested.

2/The abstract is improved according to the recommendations of the reviewer.

3/The introduction is improved in the revised version (lines 45-47).

4/the introduction is enriched on porous materials (lines 47-49 and 63-66).

5/The photo of the experimental set-up is added in the revised version.

6/Some essential points have been revised in the modified version of the conclusion.

Reviewer 4 Report

  1. In line 92, author wrote”In the high frequency domain”. What are the high frequencies here? I suggest the author add values.
  2.  Authors should add the reason why give the sensitivity analysis in the text?
  3. Maybe authors should add the corresponding references about sound absorption using JCA model. 1 Design, fabrication and sound absorption test of composite porous matamaterial with embedding I-plates into porous polyurethane[J], Applied Acoustics, 2021, 175, 107845.  2 Hybrid composite meta-porous structure for improving and broadening sound absorption[J], Mechanical Systems and Signal Processing, 154, 107504. 
  4. Please show the specific experimental photos in section 4.
  5. Please show the application of this text in the last paragraph

Author Response

We would like to thank the reviewer for the suggestions and remarks to improve our article.

The high frequency values have been added (line 101).

The reason for the sensitivity study has been added in the revised version of the article (lines 159-162).

Line 109, recent applications of the JCA model are added in the text.

The specific experimental photo is added int section 4 as requested.

Reviewer 5 Report

Review report on paper materials-1524949, entitled “Influence of higher order viscous and thermal effects on an ultrasonic wave reflected from the first interface of a porous material.”

This article concerns the study of the influence of viscous and thermal surfaces on the acoustic wave reflected by the first interface of a porous material with a rigid structure, mainly on its reflection coefficient and other parameters. Generally, when studying the acoustic wave reflection by the interface of porous materials, the viscosity and thermal surfaces are not considered. This paper not only considers the influence of the two parameters on the results but also verifies the influence through simulation and experiment. Finally, compared with traditional studies, experimental parameters such as porosity, tortuosity and characteristic lengths will change when these two parameters are considered, which provides a new research direction for subsequent studies. This reviewer believes that this work studies the theory of the acoustic wave reflected by the first interface from a deeper level so that it deserves to be published in MATERIALS. However, the following minor issues are needed to consider:

  1. There lack of some background on noise in the introduction.

Ps: The references can be used in the introduction to introduce different noise reduction methods.

[1] Optimal design of broadband quasi-perfect sound absorption of composite hybrid porous metamaterial using TLBO algorithm

[2] Design and study of a hybrid composite structure that improves electromagnetic shielding and sound absorption simultaneously

[3] Annular acoustic black holes to reduce sound radiation from cylindrical shells

[4] Transmission loss of plates with multiple embedded acoustic black holes using statistical modal energy distribution analysis

[5] Noise reduction via three types of acoustic black holes

  1. Page 1, Line 5-6, ‘limited of’-> ‘limited to’
  2. The legends in Figure 6 are different from the description of Figure 6 below. In addition, the legends presented in Figure 6 are different from the legends in other figures. For example, the green dot in other diagrams refers to the consideration of two cases, while the blue dotted line in Figure 6 represents this case.
  3. Page 11, Line 236, ‘the viscous and thermal surfaces and ’ is better to be ‘the viscous surfaces and thermal surfaces  ’.

Author Response

We would like to thank the reviewer for the suggestions and remarks to improve our article.

The five publications suggested by the reviewer are cited in the introduction in the corrected version of the article

Page 1, Line 5-6, ‘limited of’-> ‘limited to’ (this is corrected in the revised version).

Figure 6 has been changed in the corrected version of the article..

There is only one viscous surface and one thermal surface per porous material, I think we can say "viscous and thermal surfaces". If we say "viscous and thermal surfaces", it means that there are several viscous and thermal surfaces for a porous material, but this is not the case. Thank you for your suggestions.

Round 2

Reviewer 1 Report

The previous comments have been well addressed. I have two more questions.

1) Why do the incident wave and reflected wave arrive at the same time, not only in the simulation (t~1s), but also in the experiment(t~6.4s)? Usually, it should be some time of flight between the incident wave and reflected wave.

2) Although the authors claims the incident wave in the simulation is a realistic wave, where is it from? Is it analytically generated with gaussion profile with some noises or input from 5052PR? Because it is obviously different from the incident wave from experiment as Figure 11.

Besides, maybe there are too many references. Except that, I have no further comments.

Author Response

Dear Reviewer,

Thank you for your comments which improved our article and its understanding. These remarks have been taken into account.

As mentioned in line 237, a rigid metal plate is used to serve as an acoustic mirror to measure the wave emitted by the transducer (the incident wave), and since the incident wave is instantaneously reflected by the first interface of the porous material (line 60), the path traveled by the incident and reflected wave is therefore the same, which explains why the incident and reflected signal arrive at the same time. 

 The incident wave used for the simulations is taken from  our previous work published in Ref [45, 63]. This has been corrected in the new version (line 171).

I had to add some references that the reviewers asked me.

Best Regards

Z.E.A Fellah